# Gender and Psychosocial Differences in Psychological Resilience among a Community of Older Adults during the COVID-19 Pandemic

**DOI:** 10.3390/jpm12091414

**Published:** 2022-08-30

**Authors:** Alberto Sardella, Vittorio Lenzo, Giorgio Basile, Alessandro Musetti, Christian Franceschini, Maria C. Quattropani

**Affiliations:** 1Department of Clinical and Experimental Medicine, University of Messina, 98125 Messina, Italy; 2Department of Social and Educational Sciences of the Mediterranean Area, University for Foreigners “Dante Alighieri” of Reggio Calabria, 89125 Reggio Calabria, Italy; 3Department of Biomedical and Dental Sciences and Morphofunctional Imaging, University of Messina, 98125 Messina, Italy; 4Department of Humanities, Social Sciences and Cultural Industries, University of Parma, 43125 Parma, Italy; 5Department of Medicine and Surgery, University of Parma, 43125 Parma, Italy; 6Department of Educational Sciences, University of Catania, 95124 Catania, Italy

**Keywords:** clinical psychology, psychological resilience, depressive symptoms, anxiety, stress, gender differences, older adults, COVID-19 pandemic

## Abstract

The study aimed at exploring gender and additional sociodemographic differences in psychological resilience, as well as the association between resilience and psychological distress in older adults, during the first lockdown in Italy, due to the COVID-19 pandemic. Participants attended an online survey during the first lockdown in May 2020. Psychological distress was assessed through the Depression Anxiety Stress Scale-21, the Resilience Scale (RS) was administered to evaluate psychological resilience, and sociodemographic variables were also collected. The study involved 108 community older adults (mean age 70.02 ± 3.5 years). Comparisons revealed that women reported significantly lower total scores of RS (*p* = 0.027), as well as lower levels of resilience-related domains, namely Meaningfulness (*p* = 0.049), Self-Reliance (*p* = 0.011), Perseverance (*p* = 0.035), and Existential Aloneness (*p* = 0.014), compared to men. Significantly higher RS scores were found in older adults being involved in a relationship, compared to those not involved in relationships (*p* = 0.026), and in older adults with children (*p* = 0.015), compared to those without offspring, suggesting the importance for older adults of not dealing alone with such a dramatic and stressful event, such as the pandemic. Negative correlations were found between psychological resilience and stress, depression, and anxiety. Linear regressions revealed that lower RS total scores, as well as lower scores in the majority of the RS scales, were associated with greater levels of stress, greater levels of anxiety, and greater levels of depressive symptoms. This study suggested that older women might appear more vulnerable in facing the pandemic, compared to men; having not lived alone through the lockdown period might also be considered as a factor of resilience for older adults.

## 1. Introduction

The psychological impact of the COVID-19 pandemic has been increasingly recognized as a public health concern [1]. In this context, older adults appeared exposed to an increased psychological burden, especially during the experience of lockdown and its related restriction measures [2]. National governments worldwide have implemented several preventive strategies, such as quarantine and maintaining social distance, in order to contain the spread of the coronavirus and to protect the population [3]. On the other side of the coin, older adults have paid the cost of experiencing these preventive strategies in terms of social isolation, distance from relatives and friends, limited access to stimulating leisure activities, and lack of social support [4]. Such forced conditions progressively have had a negative impact on older adults’ health, by markedly affecting their cognitive and functional status, as well as their health-related quality of life [5]. In addition, the prolonged stressful experience of living during the pandemic has contributed to increasing psychological distress in the older population [6]. In line with its psychological theoretical framework, the construct of psychological distress embraces a set of multiple physical and mental symptoms, which are mainly associated with depressive symptomatology (e.g., lack of motivation, low self-esteem, and fatigue) and both somatic and subjective symptoms of anxiety. In addition, a third set of symptoms can be discriminated, namely stress, which refers to the presence of irritability, nervous tension, and agitation [7].

The pandemic, as well as its related contingencies (e.g., lockdown, quarantine, and social isolation) have affected older adults’ psychological health, exposing them to the risk of developing psychological distress. Interesting evidence has emerged from the investigation of gender differences in community populations; accordingly, women generally more than men reported higher rates of anxiety and depressive symptomatology in facing the stressful experience of the pandemic [8]. This gender difference has emerged also from studies that specifically involved older adults during the first wave of the COVID-19 pandemic. Older women have exhibited higher levels of depressive symptoms, as well as an overall worsening in mental health, compared to men [9]. Older women showed also greater (approximately doubled) odds of reporting depressive symptoms, compared to men [10]. Older women have additionally reported higher anxiety symptomatology, with negative consequences for their cognitive functioning [11]; furthermore, it has been also highlighted that older women reported higher levels of stress during the course of the pandemic, compared to men [12].

An established evidence in the literature on aging is that living alone in later life (e.g., not being involved in relationships, due to divorce or widowhood, and living without children) exposes older adults to increased psychological distress and loneliness [13,14,15]. Interestingly, it has been suggested that living alone is not a risk factor for psychological distress per se, but the transition to living alone in older age denotes a significant factor [15].

On the other side of the coin, the construct of psychological resilience has been widely described as the individual disposition to cope successfully with challenging life experiences, through the adoption of mental, emotional, and behavioral adjustment strategies to external and internal demands. In the early nineties, Gail Wagnild and Heather Young proposed a theoretical model, which was able to capture the principal characteristics of resilience and which was able to make this construct measurable [16]. According to this model, resilience was defined as a positive personality trait, which moderates the negative effects of stress, and contributes to enhanced individual adaptation to adverse outcomes and negative events. Wagnild and Young also identified five different characteristics of resilience, namely equanimity (i.e., the disposition to maintain a balanced perspective on life and experiences), perseverance (despite adversities), self-reliance (i.e., the disposition to believe in one’s capabilities and to depend on oneself), meaningfulness (i.e., the sense of having something to live for), and existential aloneness (i.e., the sense of uniqueness and freedom) [16].

In line with this theoretical framework, resilience is acknowledged as an important personal resource for older adults, by contributing to a better adjustment to stressful and negative age-related events [17,18]. Along the course of the COVID-19 pandemic, psychological resilience has been considered beneficial for older adults, especially since they needed to face several socioeconomic, psychological, and physical stressors [19]. In the general population, lower levels of resilience have been previously associated with increased psychological distress, especially in the early months of the pandemic [20]. 

Psychological resilience has been considered a factor actively involved in the adjustment to the pandemic by older adults [21]. Female gender, not being involved in relationships, and the lack of offspring have been sociodemographic factors that worsened the experience of the pandemic in older adults. The presence of different levels of psychological resilience, based on the aforementioned sociodemographic factors, is a topic not widely explored among older adults during the pandemic. The association between psychological resilience and psychological distress in older adults during the pandemic denotes a further topic of increasing interest. 

In line with these premises, the main purpose of the present study was to verify the presence of gender differences in psychological resilience among a sample of a community of older adults during the COVID-19 pandemic. The study also aimed at verifying whether older adults being involved in relationships and older adults having children showed different levels of psychological resilience, compared to those not involved in relations and with no offspring. As a secondary purpose, the study investigated the association between psychological resilience and psychological distress (i.e., depressive symptoms, anxiety, and stress) in older adults during the pandemic, by hypothesizing that these two factors were negatively associated.

## 2. Materials and Methods

### 2.1. Participants and Procedure

Data presented in this cross-sectional study resulted from a sub-analysis of a large Italian community sample, which was the object of a previous and recently published study [22]. More specifically, we selected a sample of subjects with an age ≥65 years. Participants were originally asked to voluntarily attend an online survey, during the first Italian lockdown in 2020, since the persisting of the COVID-19 pandemic-related restrictions. No form of compensation was granted. The online survey was carried out in May 2020, through the employment of the Google Modules platform [22]. Subjects who reported, via the online procedure, a positive psychiatric history were excluded from the analysis. 

### 2.2. Measures

Sociodemographic features were collected for each participant (i.e., gender, education, occupational status, marital and relationship status, and presence of offspring). 

The Italian version of the Depression Anxiety and Stress Scale (DASS-21) was employed to assess psychological distress [7,23]. The DASS-21 is a widely used 21 item self-report questionnaire, which includes a four-point Likert scale ranging from “never” (0 points) to “always” (3 points), in order to rate the frequency of specific symptoms. The questionnaire envelopes three scales, respectively assessing depressive (i.e., dysphoria, low self-esteem, anhedonia, lack of interest, and passivity), anxiety (both somatic and subjective), and stress symptoms (i.e., persistent arousal, irritability, psychological tension, and agitation). Higher scores correspond to a more severe symptomatology. According to the Lobivond and Lobivond’s cut-offs [23], the DASS-21 scores describe the symptomatology as normal, mild, moderate, severe, and extremely severe. As reported previously [22], the employed Italian version of the DASS-21 exhibited excellent levels of reliability (depression, α = 0.89; anxiety, α = 0.83; stress, α = 0.90).

Psychological resilience was assessed through the Wagnild and Young Resilience Scale (RS) [16], in its Italian version [24]. The RS is a 24 item self-report questionnaire, which includes a seven-point Likert scale from “disagree” (1 point) to “agree” (7 points), in order to rate the personal agreement with the presented statements. The questionnaire consists of five scales, which refer to specific domains of resilience. Meaningfulness evaluates the sense of having something to live for (e.g., “My life has meaning”). Self-reliance evaluates the beliefs in oneself and one’s abilities (e.g., “When I am in a difficult situation, I can usually find my way out of it”). Furthermore, perseverance measures individual perseverance despite the presence of adversity or discouragement (e.g., “Sometimes I make myself do things whether I want to or not”). Existential aloneness describes the feeling of freedom and sense of uniqueness (e.g., “I am able to depend on myself more than anyone else”). Ultimately, equanimity measures a balanced perspective vision of one’s life and experience (e.g., “I do not dwell on things that I can’t do anything about”). For the purpose of the present study, we additionally use the RS total score. According to the Italian validation [24], higher total scores express higher resilience. As previously reported [22], the five scales, and the total score, showed an acceptable-to-good reliability (self-reliance, α = 0.65; perseverance, α = 0.71; equanimity, α = 0.78; existential aloneness, α = 0.80; meaningfulness, α = 0.89; Total RS, α = 0.94). 

### 2.3. Statistical Analysis

Data were analyzed by using the software IBM SPSS Statistics version 26 (IBM Corporation, Armonk, NY, USA). Continuous variables were expressed as means and standard deviation (SD); categorical variables were expressed as count and percentages. A non-parametric approach was preferred, since most of the variables appeared not normally distributed (based on the values of skewness and kurtosis). Descriptive and correlation analyses were performed; the Mann–Whitney test for independent samples was used to test the presence of significant differences in continuous variables. Ultimately, 3-step hierarchical linear regressions were performed, in order to investigate the contribution of psychological resilience to psychological distress. We performed three regressions, by differently using the three DASS-21 scores as dependent variables. In the first step, age, education and gender were included; in the second step, relationship status, presence of offspring, and retirement status were additionally included; in the final step, the total score of the RS and the RS scales were included. 

Values of *p* < 0.05 were considered statistically significant.

## 3. Results

The study involved 108 older adults of a community (mean age 70.02 ± 3.5; 55.6% females), evaluated during the COVID-19 pandemic. The principal sociodemographic and psychological clinical characteristics of the sample are summarized in Table 1 and Table 2, respectively.

As reported in Table 2, according to the DASS-21 score’s interpretation guidelines [18,22], clinically relevant depressive symptoms were found in approximately 39% of the subjects; 25% of the subjects also reported clinically relevant anxiety symptoms. Ultimately, approximately the 22% of the subjects reported clinically relevant stress symptoms.

### 3.1. Sociodemographic Differences in Psychological Resilience

Older adults evaluated during the first lockdown reported significantly different RS scores, based on sociodemographic features. Specifically, women reported significantly lower RS total scores, and significantly lower RS scores in almost each resilience domain, compared to men (Table 3). Furthermore, older adults who were involved in relationships during the pandemic reported significantly higher RS total scores, and higher RS scores in almost each resilience domain, compared to those not involved in any relationships (Table 4). Ultimately, older adults with children reported significantly higher RS total scores, and significantly higher RS scores in almost each resilience domain, compared to those with no offspring (Table 5). 

### 3.2. Correlation Analysis, and Hierarchical Linear Regressions

The performed correlation analysis revealed that the RS total score was negatively correlated with DASS-21 Stress (*r* = −0.312; *p* < 0.001), DASS-21 Depression (*r* = −0.292; *p* < 0.001), and DASS-21 Anxiety (*r* = −0.181; *p* = 0.045); similarly, almost each RS scale was negatively correlated with the three DASS-21 scores. Correlations are extensively provided in Table 6.

Three hierarchical linear 3-step regressions were also performed, by including DASS-21 Depression, DASS-21 Anxiety, and DASS-21 Stress as the dependent variables. In the first step, age, education and gender were included; in the second step, relationship status, presence of offspring, and retirement status were additionally included; in the final step, the total score of the RS and the RS scales were included.

The final steps of the three regression models are reported in Table 7. In summary, lower RS total scores, as well as lower scores in the majority of the RS scales (except for Existential Aloneness, and partially except for Perseverance), were associated with greater levels of depressive symptoms, greater levels of anxiety, and greater levels of stress.

## 4. Discussion

The present study principally aimed at investigating the presence of different levels of psychological resilience, by comparing older adults based on sociodemographic characteristics. According to the comparisons, older women reported lower levels of psychological resilience compared to men; furthermore, greater levels of resilience were reported by older adults who were involved in affective relationships, as well as by those who could rely on the presence of sons and daughters during the pandemic. 

Older women are acknowledged to be more exposed to the occurrence of negative age-related outcomes, as well as to the negative psychological burden due to several clinical conditions [25,26]. This evidence has been dramatically confirmed even during the course of the pandemic. Consistently, older women have experienced greater levels of anxiety than men [27]; furthermore, older women reported more negative feelings and less affect balance (i.e., between positive and negative feelings) compared to men [28]. The findings of the present study appear in line with this evidence, by additionally suggesting that women’s adjustment to the stressful event of the pandemic might be affected by lower psychological resilience, compared to men.

It is known that psychological resilience predisposes individuals to good mental health despite stressor exposure. Given its psychological clinical relevance, its effective involvement during prolonged stressful events, such as the current pandemic, denotes a topic of concern. In this context, the identification of specific psychosocial and clinical factors associated with resilience has gained an increasing interest [29]. In the context of older populations, it has been previously highlighted that the active participation in advanced daily activities (e.g., hobbies) was associated with greater resilience [30]; the presence and the type of medical chronic conditions could also influence older adults’ psychological resilience [31]. The findings of the present study additionally showed that older adults being involved in a relationship, during the pandemic, exhibited greater levels of resilience; furthermore, those subjects without children exhibited lower levels of resilience. This evidence appears in line with the results of a very recent systematic review, which summarized that specific sociodemographic factors—such as education, income, network size, and social support—were correlated with older adults’ resilience, despite being weakly correlated [32]. Interestingly, the results of the present study may also reflect the importance of not dealing alone with such a dramatic and stressful event as the pandemic. These findings may be interpreted, even though indirectly, within the wider concept of loneliness, which in recent years has represented a topic of great concern from the psychogeriatric perspective [33]. Loneliness defines the perceived discrepancy between actual and desired quality (or quantity) of relationships [34], and it appears to be a psychological factor involved in aging, since older adults need to face several negative experiences and challenges, such as the lack of economic resources, the occurrence of comorbidities and disability, and the death of relatives and spouses [35]. In the light of these considerations, we can understand how living in loneliness was a common experience for many older adults during the pandemic, due to the implemented strategies aimed at containing the spread of COVID-19 (e.g., quarantine and, physical and social distancing), as well as due to the direct consequences of the disease [36]. Living in loneliness has also exposed older adults to a greater risk of developing psychological distress, and an increased risk of developing negative clinical conditions (e.g., cardiovascular and autoimmune diseases and neurocognitive disorders), as previously highlighted [37,38]. Although the present study did not specifically investigate loneliness, it appears interesting to suggest how being involved in a relationship, or having children, was associated with higher levels of psychological resilience in our sample of older adults.

As a secondary topic, the present study highlighted the presence of negative correlations between the RS scales and the DASS-21 scales, meaning that lower levels of psychological resilience were correlated with higher levels of depressive symptoms, higher anxiety levels, and higher stress symptoms. This finding appears in line with previous studies that highlighted these correlations in the general population [24,39,40]; however, few studies have verified the correlations between psychological resilience and psychological distress during the pandemic [22]. Furthermore, to the best of our knowledge, no previous studies have evidenced the presence of negative correlations between psychological resilience and psychological distress specifically in older adults, during the pandemic. The inverse association between psychological resilience and psychological distress were further confirmed by hierarchical regressions, which highlighted the contribution of higher psychological resilience to lower psychological distress. Perseverance and existential aloneness were the only two resilience dimensions that were not associated with depression, anxiety, or stress. Altogether, these results suggest that psychological resilience, and most of its dimensions, may play a significant role to explain distress in older adults. Exploring psychological resilience should be part of the routine assessment of older adults, in order to identify a potential strength to rely on. Nonetheless, these preliminary findings should be taken with caution; further studies are needed to better clarify this interesting link.

The study presents several limitations, which need to be acknowledged. First, the cross-sectional design, which does not allow us to make causal inferences, must be acknowledged. Moreover, we need to acknowledge a methodological shortcoming in the use of self-reported questionnaires, the answers in which might be affected by social desirability. An additional limitation is the lack of pre-pandemic measures of psychological distress, which might have exposed us to the risk of not accurately identify the extent of psychological distress due to the pandemic. Furthermore, the study involved a convenient, non-target sample of a community of older adults, recruited though an online survey, which might have narrowed the generalizability of the results. Furthermore, we must acknowledge the relatively small sample size, which might be explained by the potential difficulty of older adults to handle online surveys, as well as by their potential limited accessibility to smart platforms (e.g., tablet, pc, smartphone). Longitudinal studies, involving larger and targeted samples of older adults, could better clarify the long-term impact of psychological resilience dimensions on psychological distress during the pandemic. Longitudinal studies could also better confirm the impact of sociodemographic correlates of being resilient in such population.

Despite these limitations, the findings of the study appear in line with the necessity to improve the evaluation of antecedent psychological factors, which may be able to promote older adults’ adjustment to adverse outcomes and challenges [41,42]. Ultimately, similarly to psychological distress, even psychological resilience appears sensitive to sociodemographic variables, as the found differences suggested. This evidence might be considered a starting point, in order to implement tailored psychological interventions [43], which should target older women’s vulnerability, as well as older adults’ psychosocial needs.

## 5. Conclusions

Resilience might be considered a beneficial psychological contributor to reduce depression, anxiety, and stress, thus helping older adults to better face a prolonged stressful experience, such as the COVID-19 pandemic. 

In this context, older females might be considered as more vulnerable than men, since they reported lower levels of resilience. Within a psychotherapeutic perspective, older women might benefit from tailored psychological clinical interventions, in order to better cope for stressful events. Furthermore, having lived through the pandemic period without being alone might be suggested as a factor of resilience, by highlighting the positive and supportive presence of relational affections or children.

In the aging perspective, one of the challenges for the health care system is to improve personalized care strategies, in order to better capture the complexity of older adults’ health. The identification of specific differences in psychological resilience might address the implementation of personalized preventive strategies for the most vulnerable older subjects.

## Figures and Tables

**Table 1 jpm-12-01414-t001:** Main sociodemographic characteristics of the sample.

Sociodemographic Characteristics	All Sample (*N* = 108)
Age, years (mean ± SD)	70.02 (±3.53)
Gender, % (female)	60 (55.6)
Primary and secondary school, *n* (%)	9 (8.3)
High school; *n* (%)	41 (38)
Bachelor’s degree; *n* (%)	5 (4.6)
Master’s degree; *n* (%)	39 (36.1)
Postgraduate; *n* (%)	14 (13)
Married, *n* (%)	65 (60.2)
Unmarried, *n* (%)	11 (10.2)
Widow/er, *n* (%)	11 (10.2)
Divorced, *n* (%)	12 (11.1)
Other, *n* (%)	9 (8.3)
Subjects with children, *n* (%)	91 (84.3)
Retired, *n* (%)	65 (60.2)

Abbreviation: SD = Standard deviation.

**Table 2 jpm-12-01414-t002:** Psychological clinical features of the sample.

Psychological Clinical Assessment	All Sample (*N* = 108)	
Psychological Distress		
DASS-21 Depression (mean ± SD)	8.80 (±8.81)	
DASS-21 Anxiety (mean ± SD)	5.56 (±6.58)	
DASS-21 Stress (mean ± SD)	10.43 (±8.50)	
Subjects with mild to extremely severe DASS-21 Depression, *n* (%)	43 (39.8)	DASS-21 Depression (mean ± SD) = 17.40 ± 7.49
Subjects with mild to extremely severe DASS-21 Anxiety, *n* (%)	27 (25)	DASS-21 Anxiety (mean ± SD) = 14.59 ± 7.14
Subjects with mild to extremely severe DASS-21 Stress, *n* (%)	24 (22.2)	DASS-21 Stress (mean ± SD) = 23.17 ± 6.04
Psychological Resilience		
Resilience Scale (RS) total (mean ± SD)	122.19 (±36.17)	
RS Meaningfulness (mean ± SD)	35.37 (±11.37)	
RS Self - Reliance (mean ± SD)	30.62 (±8.67)	
RS Perseverance (mean ± SD)	14.61 (±4.85)	
RS Existential Aloneness (mean ± SD)	16.01 (±4.95)	
RS Equanimity (mean ± SD)	15.03 (±5.20)	

Abbreviation: SD = Standard deviation; DASS-21 = Depression Anxiety Stress Scale-21.

**Table 3 jpm-12-01414-t003:** Gender differences in psychological resilience.

Psychological Resilience	Female (*N* = 60)	Male (*N* = 48)	Effect Size	*p*
Resilience Scale (RS) total (mean ± SD)	115.35 (±41.23)	130.75 (±26.64)	0.054	**0.027**
RS Meaningfulness (mean ± SD)	33.45 (±12.98)	37.77 (±8.51)	0.024	**0.049**
RS Self - Reliance (mean ± SD)	28.75 (±9.19)	32.97 (±7.42)	0.044	**0.011**
RS Perseverance (mean ± SD)	13.73 (±5.36)	15.70 (±3.90)	0.026	**0.035**
RS Existential Aloneness (mean ± SD)	14.98 (±5.83)	17.31 (±3.17)	0.020	**0.014**
RS Equanimity (mean ± SD)	14.25 (±5.95)	16.02 (±3.90)	0.038	0.079

Note: significant *p* values (<0.05) are reported in bold.

**Table 4 jpm-12-01414-t004:** Relationship-related differences in psychological resilience.

Psychological Resilience	In a Relation (*N* = 67)	Not in A Relation (*N* = 41)	Effect Size	*p*
Resilience Scale (RS) total (mean ± SD)	130.55 (±26.82)	108.54 (±44.76)	0.046	**0.026**
RS Meaningfulness (mean ± SD)	37.80 (±8.56)	31.39 (±14.11)	0.034	0.055
RS Self - Reliance (mean ± SD)	32.32 (±7.27)	27.85 (±10.06)	0.042	**0.034**
RS Perseverance (mean ± SD)	15.73 (±3.69)	12.78 (±5.90)	0.046	**0.026**
RS Existential Aloneness (mean ± SD)	17.17 (±3.72)	14.12 (±6.06)	0.054	**0.016**
RS Equanimity (mean ± SD)	16.23 (±4.17)	13.07 (±6,10)	0.060	**0.011**

Note: significant *p* values (<0.05) are reported in bold.

**Table 5 jpm-12-01414-t005:** Offspring-related differences in psychological resilience.

Psychological Resilience	With Children (*N* = 91)	No Children (*N* = 17)	Effect Size	*p*
Resilience Scale (RS) total (mean ± SD)	125.55 (±35.17)	104.24 (±37.18)	0.055	**0.015**
RS Meaningfulness (mean ± SD)	36.39 (±11.22)	29.88 (±10.91)	0.062	**0.010**
RS Self - Reliance (mean ± SD)	31.45 (±8.38)	26.23 (±9.16)	0.045	**0.028**
RS Perseverance (mean ± SD)	14.96 (±4.82)	12.70 (±4.70)	0.036	0.051
RS Existential Aloneness (mean ± SD)	16.41 (±4.82)	13.88 (±5.20)	0.055	**0.015**
RS Equanimity (mean ± SD)	15.43 (±5.11)	12.88 (±5.26)	0.038	**0.041**

Note: significant *p* values (<0.05) are reported in bold.

**Table 6 jpm-12-01414-t006:** Correlations between DASS-21 and RS scales.

	1	2	3	4	5	6	7	8	9
1. DASS-21 Depression	1								
2. DASS-21 Anxiety	0.609 **	1							
3. DASS-21 Stress	0.747 **	0.687 **	1						
4. Resilience Scale (RS) total	−0.292 **	−0.181 *	−0.312 **	1					
5. RS Meaningfulness	−0.286 **	−0.199 *	−0.315 **	0.955 **	1				
6. RS Self-Reliance	−0.226 *	−0.128	−0.255 **	0.928 **	0.837 **	1			
7. RS Perseverance	−0.288 **	−0.151	−0.313 **	0.904 **	0.865 **	0.815 **	1		
8. RS Existential Aloneness	−0.167	−0.151	−0.218 *	0.843 **	0.811 **	0.753 **	0.714 **	1	
9. RS Equanimity	−0.246 *	−0.238 *	−0.288 **	0.844 **	0.780 **	0.720 **	0.751 **	0.790 **	1

** = *p* value ≤ 0.001; * = *p* value ≤ 0.05.

**Table 7 jpm-12-01414-t007:** Linear regressions for DASS-21 Depression, DASS-21 Anxiety and DASS-21 Stress.

Dependent Variables	Independent Variables	Model		Coefficients			
		R^2^	F				
DASS-21 Depression		0.248 *	2.617				
				SE(B)	*β*	*t*	*p*
	Age			0.240	−0.054	−0.555	0.580
	Gender			1.766	0.054	0.537	0.593
	Education			0.680	0.016	0.161	0.872
	Relationship status			1.882	−0.098	−0.937	0.351
	Presence of children			2.372	0.033	0.340	0.735
	Retirement status			1.892	−0.022	−0.213	0.832
	RS Total score			0.449	−2.734	2.028	**0.045**
	RS Meaningfulness			0.610	−1.707	−2.135	**0.035**
	RS Self - Reliance			0.516	−1.145	−2.155	**0.026**
	RS Perseverance			0.614	−0.733	−2.171	**0.032**
	RS Existential Aloneness			0.581	−0.314	0.962	0.339
	RS Equanimity			0.563	−0.878	−1.644	**0.010**
		R^2^	F				
DASS-21 Anxiety		0.226 *	2.314				
				SE(B)	*β*	*t*	*p*
	Age			0.182	−0.008	−0.081	0.936
	Gender			1.338	0.011	0.112	0.911
	Education			0.515	−0.005	−0.050	0.960
	Relationship status			1.426	−0.156	−1.480	0.142
	Presence of sons			1.797	−0.016	−0.158	0.875
	Retirement status			1.434	−0.021	−0.196	0.845
	RS Total score			0.340	−4.758	2.547	**0.012**
	RS Meaningfulness			0.469	−2.224	−2.741	**0.007**
	RS Self - Reliance			0.391	−1.154	−2.240	**0.027**
	RS Perseverance			0.465	−0.498	−1.454	0.149
	RS Existential Aloneness			0.441	−0.019	−0.056	0.955
	RS Equanimity			0.426	−1.339	−3.973	**<0.001**
		R^2^	F				
DASS-21 Stress		0.267 *	2.881				
				SE(B)	*β*	*t*	*p*
	Age			0.229	−0.149	−1.546	0.125
	Gender			1.683	0.052	0.529	0.598
	Education			0.648	0.012	0.123	0.902
	Relationship status			1.793	−0.134	−1.302	0.196
	Presence of sons			2.260	−0.035	-0.356	0.722
	Retirement status			1.803	−0.006	−0.059	0.953
	RS Total score			0.427	−3.367	1.852	**0.047**
	RS Meaningfulness			0.590	−1.692	−2.143	**0.035**
	RS Self - Reliance			0.491	−1.042	−2.077	**0.041**
	RS Perseverance			0.585	−0.577	−1,730	0.087
	RS Existential Aloneness			0.554	−0.351	1.089	0.279
	RS Equanimity			0.536	−0.842	−2.567	**0.012**

Notes: significant *p* values (<0.05) are reported in bold. * model *p* ≤ 0.05.

## Data Availability

The raw data supporting the conclusions of this article will be made available by the authors, without undue reservation.

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
