# Peer review of "Gender and Psychosocial Differences in Psychological Resilience among a Community of Older Adults during the COVID-19 Pandemic"

_jpm, 2022, doi:10.3390/jpm12091414_

Round 1

Reviewer 1 Report

Thank you for the opportunity to review an article that deals with an important and current topic. The strength of the article is its clear structure, orderly presentation, and brevity. The manuscript, however, has many disadvantages and requires a lot of improvement. Starting with the introduction, it is necessary to precisely conceptualize the concepts of stress, distress, depression, anxiety, and psychological resilience in the sense of personal resources. In the introduction, the authors also omit sociodemographic variables significant in the analysis and the results obtained concerning having children and being in a relationship. The aim of the study does not coincide with the content of the presented results.
The description of the measurement tools lacks information on RS's reliability level. Where did the researchers get scores of 126.6 as a high level of resilience? What is the theoretical basis of this measurement tool? It should be described in the introduction.
In table 1 mean age is 70.02, and in text, 68.70. What does it mean in table 2: DASS-21 Depression /Anxiety / Stress and Symptomatic DASS-21 1 Depression /Anxiety / Stress? What is the difference? It is unclear.
A large part of the article is devoted to elementary statistics, such as descriptive statistics, intergroup comparisons, and correlations (included as a supplement to the article).
The authors dedicated 4 lines to regression analysis and described it briefly and incorrectly. They did not include any table. The most advanced manuscript analysis was neglected by the researchers. This is the biggest objection.
The discussion requires improvement and deepening of conclusions based on the obtained results. The differences between gender in the results should be considered in the literature and potential psychological and sociological mechanisms. Much of the discussion focuses on loneliness, which is not referred to in the introduction. The problem of interpersonal relations of older women and men and their impact on mental health has been omitted in the introduction and over-emphasized in the discussion. The limitations of the research should be more specific and point to other real shortcomings (the studied group as a non-target group, conceptual difficulties related to measurement tools, analysis of simple dependencies).
The discussion includes overinterpreting sentences (for example, lines 187-190). First, it is necessary to organize the terminology: mental health, stress, depression, anxiety, distress, adaptation, resilience. These concepts are close in meaning but not identical. Moreover, conclusions from the intergroup comparisons cannot be attributed to findings from regression or indirect analyses that have not been performed.
Summing up, the idea seems interesting, but the article requires improvement of the whole: Theoretical in-depth, a different presentation of the results (perhaps indirect relationships between the variables or logistic regression with an indication of risk factors; maybe it would be interesting to look at the components of resilience, their correlations with depression, anxiety, stress or their lack, referring to the results already presented (page 5), and showing in discussion). It seems that it will be necessary for the authors to explore the resilience/resiliency construct literature, which will allow them to understand the obtained results and plan a more advanced analysis.

Author Response

We gratefully thank the Reviewer for the positive opinion on the study. We do hope to have welcomed each suggestion, and to have improved the overall quality of the manuscript. Revisions were highlighted in yellow in the revised version of the manuscript.

Reviewer 2 Report

This manuscript investigates gender end psychosocial difference in psychological resilience of older adults at the time of the Covid-19 pandemic. This is an interesting and relevant topic, given that this pandemic is still ongoing, and it is therefore important to know which factors promote resilience against the impact of the pandemic and which groups reveal a greater psychosocial vulnerability with regard to the pandemic and its consequences.

However, I have several comments and concerns:

-          I think a crucial limitation of this study is its cross-sectional design. In consequence, we cannot tell how depression, anxiety, stress, and resilience have changed due to the pandemic, as there is no pre-pandemic measurement occasion available. Also, it is unclear to me if the associations, e.g. between gender and resilience, are really “pandemic-specific” or if they rather correspond to the associations which were already found prior to the pandemic. The authors should address such limitations. They also might want to compare their empirical prevalence rates of depression and anxiety with normative values from before the pandemic, so that we can get an impression whether the prevalence could have changed due to Covid-19. Same regarding the association of gender and other factors with resilience

-          I missed a more conceptual and theoretical perspective on resilience. Which are the main theories of resilience? What is already known about antecedents of resilience? It is unclear why the authors focused on gender, relationship status and “having sons” (why only sons? What about daughters, and what about social networks in general?) as correlates of resilience – I believe there might be other factors that are more closely related with resilience. Why would we expect gender differences in resilience, what are the underlying mechanisms/reasons for such a gender difference?

-          Intro: The introduction emphasizes the role of older adults as a risk group with regard to negative psychological consequences of the pandemic. In their own study, they do not have a younger comparison group available (another study limitation) so that this issue cannot be further investigated. However, the authors should be careful and avoid ageism. Expressions such as “the weakest populations as older adults” should be removed. As far as I know the literature, the negative psychosocial consequences of the pandemic actually seem to be more severe in younger or middle-aged than in older adults, and this is an important issue that the authors may want to point out and discuss as well. The socio-economic stressors which the authors mention on p. 2 may be more severe for those (younger/middle-aged) adults who are still in the workforce, whereas retired older adults may not be strongly socioeconomically affected by the pandemic.

Röhr, S., Reininghaus, U., & Riedel-Heller, S. G. (2020). Mental wellbeing in the German old age population largely unaltered during COVID-19 lockdown: results of a representative survey. BMC geriatrics, 20(1), 489. https://doi.org/10.1186/s12877-020-01889-x

Bäuerle, A., Teufel, M., Musche, V., Weismüller, B., Kohler, H., Hetkamp, M., ... Skoda, E.-M. (2020). Increased generalized anxiety, depression and distress during the COVID-19 pandemic: a cross-sectional study in Germany. Journal of Public Health. https://dx.doi.org/10.1093/pubmed/ fdaa106

Benke, C., Autenrieth, L. K., Asselmann, E., & Pané-Farré, C. A. (2020). Lockdown, quarantine measures, and social distancing: Associations with depression, anxiety and distress at the beginning of the COVID-19 pandemic among adults from Germany. Psychiatry Research, 293, 113462. https://doi.org/10.1016/j.psychres.2020.113462

Majse Lind, PhD, Susan Bluck, PhD, Dan P McAdams, PhD, More Vulnerable? The Life Story Approach Highlights Older People’s Potential for Strength During the Pandemic, The Journals of Gerontology: Series B, Volume 76, Issue 2, February 2021, Pages e45–e48

-          In the introduction, the authors point out that during the pandemic, women have higher levels of depression than men; however, to what extent is this really due to the pandemic? Depressive symptoms seem to be generally more pronounced in women than in men, see for instance: Sutin et al. (2013) The Trajectory of Depressive Symptoms Across the Adult Life Span

-          Method:

-          Please provide exemplary items for the different scales (e.g. resilience and resilience subscales); also psychometric properties (Cronbach’s alpha) for the resilience scale need to be reported

-          What was the reason for carrying out Friedman’s test and Wilcoxon test, rather than parametric tests? The tests chosen might come with lower statistical power.

-          Why was only the total resilience score used for predicting stress, anxiety, and depression, rather than the resilience subscales?

-          Results:

-          Tables 3-5: a column displaying effect sizes should be added in each of these tables

-          Regression findings are only reported, but I missed a table showing all regression coefficients as well as R²

-          Discussion:

-          I think another study limitation is that all measures were self-reported; influences such as social desirability could thus play a role

-          In the discussion, the authors refer to “adaptation to age-related negative outcomes and challenges” (p. 6). I am not sure what is meant here. Would we interpret the pandemic as an “age-related challenge”, or isn’t the pandemic an example for a rather “age-independent” challenge, requiring adaptation efforts from all age groups?

-          The authors refer to loneliness, but what they assessed is partnership status (which is not necessarily loneliness); generally, some of the conclusions do not seem to match with the actual set of study variables that were available and analyzed.

-          Minor issues:

-          Abstract: It would be good to state the country where study participants were recruited already here, as different countries were to a different extent hit by the pandemic. Also, the time of assessment should be reported. “Related subjects” sounds a bit odd to me; rather “subjects who lived in a relationship”?

-          P. 2 “…have resulted a consistent finding” – this sounds awkward (and grammatically wrong)

-          P. 2 “coping for challenging life experiences” – should be changed to “coping with…”; I think a thorough proof-reading of all parts of the manuscript is actually needed

-          Table 2 “meaninguflness” (typo; also in Table 3)

-          P. 5 “then men” needs to be changed to “than men”

Author Response

We gratefully thank the Reviewer for the positive opinion on the study. We do hope to have welcomed each suggestion, and to have improved the overall quality of the manuscript. Revisions were highlighted in yellow in the revised version of the manuscript
